# A machine learning model of microscopic agglutination test for diagnosis of leptospirosis

Yuji Oyamada[1], Ryo Ozuru[2,¤]*, Toshiyuki Masuzawa[3], Satoshi Miyahara[4], Yasuhiko Nikaido[4], Fumiko Obata[2], Mitsumasa Saito[4], Sharon Yvette Angelina M. Villanueva[5], Jun Fujii[2]

1 Department of Electrical Engineering and Computer Science, Faculty of Engineering, Tottori University, Tottori, Japan, 2 Division of Bacteriology, Department of Microbiology and Immunology, Faculty of Medicine, Tottori University, Yonago, Tottori, Japan, 3 Laboratory of Microbiology and Immunology, Faculty of Pharmaceutical Sciences, Chiba Institute of Science, Choshi, Chiba, Japan, 4 Department of Microbiology, School of Medicine, University of Occupational and Environmental Health, Kitakyushu, Fukuoka, Japan, 5 Department of Medical Microbiology, College of Public Health, University of the Philippines Manila, Manilla, Philippines

☯ These authors contributed equally to this work.
¤ Current address: Department of Microbiology and Immunology, Faculty of Medicine, Fukuoka University, Fukuoka, Japan
* ozuru@fukuoka-u.ac.jp

**Data Availability Statement:** All relevant data are within the manuscript and its Supporting Information files.

## Abstract

Leptospirosis is a zoonosis caused by the pathogenic bacterium *Leptospira*. The Microscopic Agglutination Test (MAT) is widely used as the gold standard for diagnosis of leptospirosis. In this method, diluted patient serum is mixed with serotype-determined Leptospires, and the presence or absence of aggregation is determined under a dark-field microscope to calculate the antibody titer. Problems of the current MAT method are 1) a requirement of examining many specimens per sample, and 2) a need of distinguishing contaminants from true aggregates to accurately identify positivity. Therefore, increasing efficiency and accuracy are the key to refine MAT. It is possible to achieve efficiency and standardize accuracy at the same time by automating the decision-making process. In this study, we built an automatic identification algorithm of MAT using a machine learning method to determine agglutination within microscopic images. The machine learned the features from 316 positive and 230 negative MAT images created with sera of *Leptospira*-infected (positive) and non-infected (negative) hamsters, respectively. In addition to the acquired original images, wavelet-transformed images were also considered as features. We utilized a support vector machine (SVM) as a proposed decision method. We validated the trained SVMs with 210 positive and 154 negative images. When the features were obtained from original or wavelet-transformed images, all negative images were misjudged as positive, and the classification performance was very low with sensitivity of 1 and specificity of 0. In contrast, when the histograms of wavelet coefficients were used as features, the performance was greatly improved with sensitivity of 0.99 and specificity of 0.99. We confirmed that the current algorithm judges the positive or negative of agglutinations in MAT images and gives the further possibility of automatizing MAT procedure.

**Funding:** This work was partially supported by JSPS KAKENHI Grant Number 18K16174 and 21K16320 to R.O., the discretionary fund of Tottori University President to Y.O. and R.O, and the Research Program of the International Platform for Dryland Research and Education, Tottori University to J.F. The funders had no role in study design, data collection and analysis, decision to publish, or preparation of the manuscript. There was no additional external funding received for this study.

**Competing interests:** The authors have declared that no competing interests exist.

# Introduction

Leptospirosis, an infectious disease caused by the pathogenic species of *Leptospira*, is one of the most widespread zoonoses in the world. The World Health Organization (WHO) estimates one million leptospirosis cases and 58,900 deaths worldwide each year, of which more than 70% is occurring in the tropical regions of the world [1]. Nonspecific and diverse clinical manifestations make clinical diagnosis difficult, and it is easily misdiagnosed with many other diseases in the tropics, such as dengue fever, malaria, and scrub typhus [2].

Microscopic agglutination test (MAT) is considered as the standard test for serological diagnosis of leptospirosis [3]. Leptospires have over 250 serovars [4] and MAT is usually used to diagnose patients based on the *Leptospira* serotypes that infect humans or animals. The principle of MAT is simple, it consists of mixing the serially diluted test serum with a culture of leptospires and then evaluating the degree of agglutination due to immunoreaction using a dark-field microscope [2]. The highest serum dilution which agglutinates 50% or more of the leptospires is considered to be the antibody titer. However, the procedures involved in MAT, especially judging the results (*i.e.*, whether positive or negative) requires highly trained personnel, thus making it difficult to adopt as a general test [5]. Furthermore, the liquid handling such as transferring all the samples from each well of multi plates onto slide glasses is complicated and time consuming. Although the International Leptospirosis Society has been implementing the International Proficiency Testing Scheme for MAT for several years now [6], worldwide standardization of MAT is yet to be achieved. This is because not only devices used for MAT (dark-field microscopes, objective lenses, illuminations and cameras) vary, but also the testing condition (the dilution range of serum, incubation time and magnification of an objective lens used) of MAT is diverse among various laboratories.

Clinicians have utilized several machine learning techniques for this kind of infectious disease diagnosis [7] and non-infectious diseases, such as cancers [8]. Focusing on image analysis, one of the most successful examples of machine learning methods in diagnosis is the classification of skin cancer [9]. In the machine learning model in the skin cancer study, a dermatologist-labelled dataset of 129,450 clinical images from 2032 different diseases was used for training. This well-trained model was able to classify skin cancers at the "dermatologist level". Since diagnosis by clinicians such as pathologists is not only based on visual inspection of the lesion but on a variety of factors, these machine learning techniques do not enable every diagnosis in clinical practice. However, patients will gain benefit by machine learning models that accurately classify images at the clinician level, even in areas where hospitals and clinicians are limited.

In theory, most disease diagnoses are regarded as a pattern classification problem [10]. Given input medical image(s), a pattern classification method first extracts image features from the input image(s) and then categorizes the extracted feature into one of predefined classes, positive or negative for simple decision and multiple classes for severity decision (see S1 File for binary classification which explains "positive or negative decision"). The key factors of successful classification are the feature extraction (feature engineering) and categorization (classifiers). Feature engineering designs an image feature and its extraction procedure such that the extracted feature well-represents the characteristics of the input images [11]. Classifiers are supposed to categorize the extracted features into their appropriate categories, infected ones as positive and non-infected ones as negative, and machine learning techniques are used to tune classifiers for maximum performance; this is called training [12].

When we design a new decision support system for infectious disease diagnosis, the key factor, feature engineering, requires knowledge and expertise on the disease while the other key factor, pattern classification, requires machine learning expertise that is often publicly available

as (open source) software. One of intuitive examples is feature engineering for melanoma diagnosis, which is conducted based on the Asymmetry, Border irregularity, Color variegation, Diameter (ABCD) rule [13]. The task of feature engineering for this is to extract ABCD related features from dermoscopic images that contain image segmentation and shape analysis [14].

For binary classification on images, we use image features instead of raw images, that are considered too complex, redundant and potentially badly distributed data. An image feature, extracted from an image, encodes characteristic information of the image into numerical values such as vector and matrix [15]. Appropriate image features for binary classification are sensitive to class differences but not to other factors, such that negative and positive data are distributed separately. Therefore, it is important to use or design appropriate image features for the target problem.

In this paper, we developed a model to verify the possibility of applying a machine learning technique to MAT. This is the first report on machine learning techniques for MAT diagnosis that uses a binary classification where each MAT image is taken either as agglutination positive or negative. This study aims to: (1) establish a machine learning based method for reading MAT results to aid in serodiagnosis; (2) design a MAT image feature that is appropriate for binary classification based on general machine learning and image processing techniques; and (3) conduct an evaluation test with MAT images obtained from animal experiments. This study will be the first step of our ultimate goal that is to fully automate the MAT procedure.

## Materials and methods

This section describes the proposed method that uses machine learning techniques for MAT image classification which predicts each MAT image either as agglutination positive or negative.

Fig 1 shows the complete training and test pipeline. All the MAT images obtained through animal experiments are pre-processed and stored in the database. The proposed method first

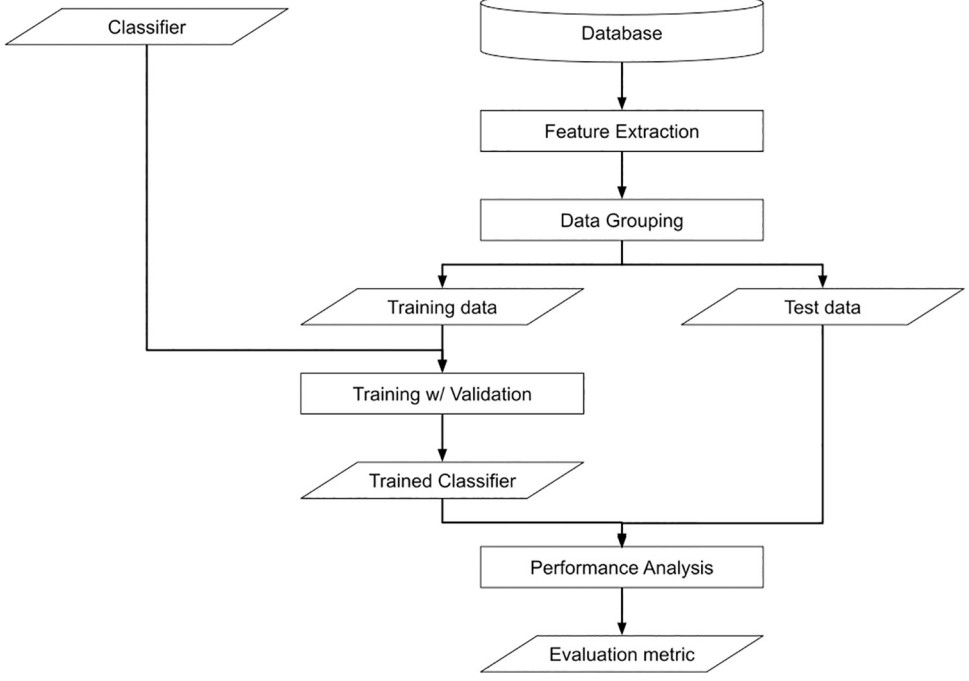

**Fig 1. The flowchart of the proposed method.**

**Table 1. The specification of raw MAT images.**

|  | #Data | Resolution [pixels] | Scale [μm/pixel] |
|---|---|---|---|
| Negative | 32 | 2448 × 1920 | 5.8 |
| Positive | 263 | 2448 × 1920 | 11.6 |

extracts image features from each patch (a small image of the original image divided by a particular resolution), and the extracted features are separated as either training or test data. With training data, the proposed method tunes the classifier in a cross-validation manner. We measure the performance of the trained classifier with test data.

## Animal ethics statement

Animal experiments were reviewed and approved by the Ethics Committee on Animal Experiments at the University of Occupational and Environmental Health, Japan (Permit Number: AE15-019). The experiments were carried out under the conditions indicated in the Regulations for Animal Experiments of the university and Law 105 and Notification 6 of the Government of Japan.

## Preparation of image datasets of MAT

In this study, a total of three golden Syrian hamsters, male, 3-week-old were used (purchased from Japan SLC, Inc., Shizuoka, Japan). Serum from a hamster subcutaneously infected with $10^4$ *Leptospira interrogans* serovar Manilae at day 7 post infection was used as a MAT positive sample, and sera from other two hamsters at day 0 post infection was used as a MAT negative. MAT was performed according to the standard manual [2]. In brief, each serum was primarily diluted 50-fold and then serially diluted (2-fold) until 25,600-fold dilution. Leptospires $(1 \times 10^8)$ were added to each diluted serum and incubated at 30˚C for 2–4 hours. Samples from each well were transferred to glass slides and covered with coverslips. Each sample was observed under a dark-field microscope (OPTIPHOT, Nikon, Tokyo Japan). Ten images from 20× or 40× objective lens fields per slide were obtained with a CCD camera (DP21, Olympus, Tokyo, Japan).

We applied some image processing to all the MAT images to make them under similar lighting conditions and the same meter-resolution scale. Tables 1 and 2 show the data properties. We have a total of 295 images, which consist of 32 negative and 263 positive images. All images are in a resolution of 2448 × 1920. The micrometer-pixel ratio is 5.8 and 11.6 in negative and positive images, respectively. We resized all positive data so that the micrometer-pixel ratio is consistent across the classes. For the detail of this pre-processing, the readers can refer to the S1 File.

## Feature extraction

In this study, we considered the following three image features: raw images (*Image*), multi-level wavelet coefficients (*Wavelet*), and the histogram of multi-level wavelet coefficients (*HoW*) (Fig 2). There was still difficulty in the pre-processed MAT image patches such as

**Table 2. The specification of scale normalized MAT images.**

|  | #Data | Resolution [pixels] | Scale [μm/pixel] |
|---|---|---|---|
| Negative | 32 | 2448 × 1920 | 5.8 |
| Positive | 263 | 1224 × 960 | 5.8 |

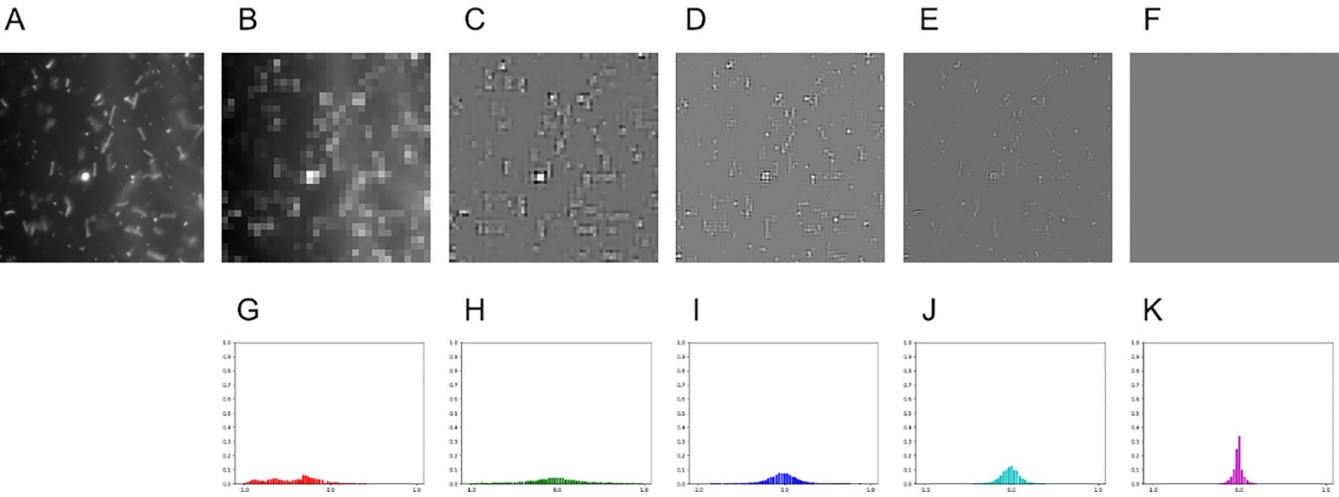

**Fig 2. An image and its image features.** (A) An image. (B)-(F) The 0-th to 4-th level wavelet coefficients of (A) (*Wavelet-0, . . ., Wavelet-4*). (G)-(K) The histogram of the 0-th to 4-th level wavelet coefficients of (A) (*HoW-0, . . ., HoW-4*).

images that might contain non-agglutination objects as dust with strong reflection. Consequently, non-agglutination objects from the pre-processed MAT image patches are excluded and the occupancy of the agglutination area is measured. The designed image feature is the histogram of multi-level wavelet coefficients of a MAT image patch. The multi-level wavelet transform [16] was used to extract objects on MAT image patches that have similar size to agglutination. By using the histogram, the behavior of the coefficients can be efficiently represented (Fig 2G–2K).

**Raw images (*Image*).**   One of the simplest image features is raw image itself. A gray-scale image is represented as a 2D matrix $I \in R^{X \times Y}$, where $X$ and $Y$ denote the image width and height. A pixel intensity at pixel $(x, y)$ is represented as $I_{x,y}$ and the value is usually expressed by an eight-bit integer, 0, 1, 2, . . ., 255. When raw images are used as an image feature, comparison between two images was done by comparing their corresponding pixel intensities. For instance, mean square error between two images $I$ and $I'$ is defined as

$$E_{MSE} = \frac{1}{XY} \sum_{y=1}^{Y} \sum_{x=1}^{X} \left( I_{x,y} - I'_{x,y} \right)^2$$

Raw images are easy to use and require less computer power. However, the pixel-wise comparison is inappropriate to consider the occupancy of the agglutination areas.

**Multi-level wavelet coefficients (*Wavelet*).**   We apply multi-level wavelet transformation to extract agglutination of a certain size from MAT images while excluding other objects. The idea of multi-level wavelet transformation [16], which decomposes an image to a combination of a base signal with different directions and resolutions, is similar to the Fourier transformation. Applying multi-level wavelet transformation to a MAT image patch, any objects on the patch are separately extracted and stored into coefficients at different scales as larger coefficients. The lowest scale coefficients $W_0 \in R^{X_l \times Y_l}$ is the mean image of the patch while the other coefficients $\{W_l \in R^{X_l \times Y_l} | l = 1, \ldots, L\}$ contain large values where image objects of specific size exist at the location, where $X_l$ and $Y_l$ denote the width and height of l-th level as

$$X_l = \frac{X}{2^{(L-l+1)}}$$

and

$$Y_l = \frac{Y}{2^{(L-l+1)}}$$

Lower-level coefficients have the information of smaller image objects and higher-level coefficients have one of larger image objects. In other words, lower-level coefficients correspond to coarser image components such as rough shape objects and larger level coefficients to finer components.

Fig 2A–2F show same microscopic field in various multi-level wavelet coefficients with Haar wavelet as base signal and 4 levels. As shown in the figure, image objects of different sizes are extracted in different levels, that coarser features are in lower-level and finer ones are in higher. In these figures, zero coefficients are depicted as gray pixels and strong features such as points, curves, and edges with large coefficient changes are depicted as white and black pixels. Trained MAT examiners confirmed that *Leptospira*-like objects showed brighter pixels in levels 2 and 3 but not in the other levels. Multi-level wavelet transformation to MAT image patches was applied and decided their coefficients of a level might be a candidate of the image feature. Benefit of this feature is that we can exclude non-agglutination objects based on their size. The drawback is the same as the *Image* feature, that *Wavelet* feature also applies pixel-wise comparison.

**Histogram of multi-level wavelet coefficients (*HoW*).** We constructed a normalized histogram of multi-level wavelet coefficients per level with a sum of 1 to represent the occupancy of the agglutination areas in a MAT image. Constructing the histogram of wavelet coefficients is equivalent to count agglutination-like objects and therefore it is appropriate to measure the occupancy of the agglutinated areas. Multi-level wavelet transformation was first applied and the histogram of the coefficients per level was constructed. Next, the histogram with a sum of 1 was normalized. By normalizing the histogram, different image resolutions are comparable. In total, we have a set of $L+1$ histograms as $\{H_l \in R^B | l = 0,\ldots,L\}$ where $H_l$ denotes the normalized histogram of $l$-th level wavelet coefficients and $B$ denotes the number of histogram bins. Benefits of this feature are its robustness against the existence of non-agglutination objects.

Fig 2G–2K show *HoW*s at different levels of an image. As shown in the figure, histograms at different levels have different sharpness. Since each level of wavelet coefficients has zero at most of the pixels, their histogram has their peak at the center. When the patch contains image objects of a specific size, their corresponding histograms have gentle peaks.

## Classifier

Here, we explain the details of binary classification. The proposed method utilizes the Support Vector Machine (SVM) [17], which is one of the supervised learning models for classification problems and has been well-used in practical situations because of its generalization performance against unknown data. Moreover, grid search hyper-parameter optimization and K-fold cross validation were combined in order to obtain better training effects.

**SVMs.** The proposed method utilized Support Vector Machines (SVMs) [17], which is one of the well-used supervised learning models for solving classification problems. Suppose we have a set of $N$ training data $D = \{(x_i, y_i) | x_i \in R^M, y_i \in R^M, y_i \in \{-1,1\} | i = 1,\ldots,N\}$, where $x_i$ denotes feature vector and $y_i$ denotes its label. The label $y_i = -1$ indicates $i$-th feature is negative data and $y_i = 1$ indicates it is positive data. In our case, $x_i$ is a MAT image feature, and negative and positive data means non-agglutinated and agglutinated data, respectively. Given the set of training data $D$, an SVM finds a boundary that maximizes its margin, which is the largest distance to the nearest data of both classes.

Using a kernel trick, SVMs accomplish even non-linear classification. Using a kernel function, a SVM alters original data $x_i$ to higher dimensional space, and linearly separates the calculated data in the projected domain. The kernel function decides the complexity of boundary shape. Typical kernel functions for non-linear classifications are polynomial functions

$$k_{poly}(x_i, x_j) = \gamma(x_i \cdot x_j)^d,$$

where $\gamma$ denotes the scale factor and $d$ the dimensionality of the polynomial and radial basis functions

$$k_{rbf}(x_i, x_j) = exp(-\gamma\|x_i - x_j\|^2),$$

where $\gamma$ denotes the non-negative scale factor.

**Grid search hyper-parameter tuning.**   The proposed method utilized grid search for tuning hyperparameters of SVMs. The hyperparameters of SVMs are parameters of kernel function, e.g., $\gamma$ of polynomial and radial basis functions, and general SVM parameter $C$. Grid search is one of the hyperparameter tuning methods that exhaustively considers all potential combinations of hyperparameters.

## Results

We conducted three evaluations to validate the potential of machine learning and image processing techniques on MAT: (1) the elapsed time of the feature extraction process, (2) qualitative evaluation of the MAT image feature classification, and (3) quantitative evaluation of the MAT image feature classification.

### Experimental conditions

Here, we describe the experimental conditions. We conducted all the experiments on a computer with an 8-core central processing unit (CPU) (Intel(R) Xeon(R) CPU E5-2620 v4) and 128 GB random access memory (RAM). All the data is stored on a hard disk drive (HDD). We implemented the proposed method using Python with standard packages such as numpy and scikit-learn, and an open-source package "PyWavelet" with Multi-level Wavelet Transformation [18].

In these experiments, we tested two patch sizes $256 \times 256$ and $512 \times 512$. The number of patches for each patch size is shown in Table 3.

For image features, $L$, the level of multi-level wavelet transformation, is set to 4, whereas $B$, the number of histogram bins, is set to 64. We tested image features: *Image*, *Wavelet* and *HoW* features at each level, denoted as *Wavelet-l* and *HoW-l*. Moreover, we combined all levels of *Wavelet* and *HoW*, denoted as combined *Wavelet* and combined *HoW*, and tested as well.

Table 4 shows the dimensionality of each feature with different patch sizes. *Wavelet-l* dimensionality becomes larger as the level increases, while *HoW-l* has constant dimensionality.

### Elapsed time of feature extraction

The first experiment measures the elapsed time of the feature extraction process. Fig 3 visualizes the elapsed time of the image processing steps. Note that the pre-processing is executed

**Table 3. The specification of scale normalized MAT images.**

|  | $256 \times 256$ | $512 \times 512$ |
| --- | --- | --- |
| Negative | 1680 | 360 |
| Positive | 2688 | 448 |
| Total | 4368 | 808 |

**Table 4. The dimensionality of the image features.**

|  | 256 × 256 | 512 × 512 |
|---|---|---|
| *Image* | 65536 | 262144 |
| *Combined-Wavelet* | 65536 | 262144 |
| *Wavelet-0* | 256 | 1024 |
| *Wavelet-1* | 768 | 3072 |
| *Wavelet-2* | 3072 | 12288 |
| *Wavelet-3* | 12288 | 49152 |
| *Wavelet-4* | 49152 | 196608 |
| *Combined-HoW* | 320 | 320 |
| *HoW-0* | 64 | 64 |
| *HoW-1* | 64 | 64 |
| *HoW-2* | 64 | 64 |
| *HoW-3* | 64 | 64 |
| *HoW-4* | 64 | 64 |

per raw image while the feature extraction is executed per patch. Thus, patch size affects the elapsed time only on the feature extraction. The pre-processing is roughly 4 Hz that is far from real-time applications that require at least 30 Hz. The feature extraction is dominated by Wavelet transformation; however total elapsed time for *HoW* computation is roughly 60 Hz even for the larger patch size.

To improve the computational speed, we can use both hardware and software level techniques. The hardware technique is to use solid state drive (SSD) instead of HDD so that the process *Image Load* becomes faster. One of potential software techniques is parallel programming that simultaneously processes more than a single image such as multi-core programming and graphics processing unit (GPU) programming. To improve and globalize the computing environment, a potential application can be a cloud-based system, in which users send raw MAT images via the internet and all the processes are performed on the server PC with multi-core CPU and GPU.

## Qualitative evaluation of MAT image feature classification

The second experiment is a qualitative evaluation of the MAT image feature classification. In this experiment, we visualize the distribution of all datasets in each image feature domain to see which image feature is suitable for image classification.

For this evaluation, we use T-distributed Stochastic Neighbor Embedding (t-SNE) [19]. In t-SNE, a non-linear dimensionality reduction method, it embeds high dimensional data into lower dimensions of two or three dimensions in such a way that similar data are distributed closer and dissimilar data are distributed further in the lower dimension with high probability. When t-SNE embeds image features of each class into distinctly isolated clusters, the features have potential ability to be good features for classification.

We set the hyper-parameter of t-SNE as follows: perplexity is 50 and the number of iterations is 3000. We applied t-SNE visualization for all combinations of image features and resolutions of 256 × 256 and 512 × 512.

Fig 4 compares t-SNE visualization of each image feature. Here, we show *Image*, *Wavelet-3*, and *HoW-3* of the resolution 256 × 256. For the remaining plots, the readers can refer to the S1 File. In each plot, red (x) and purple (+) symbols represent negative and positive data, respectively, and values of both axes are omitted because the scale does not matter in the embedded lower dimensional spaces. As shown in the figure, both negative and positive data of *Image*

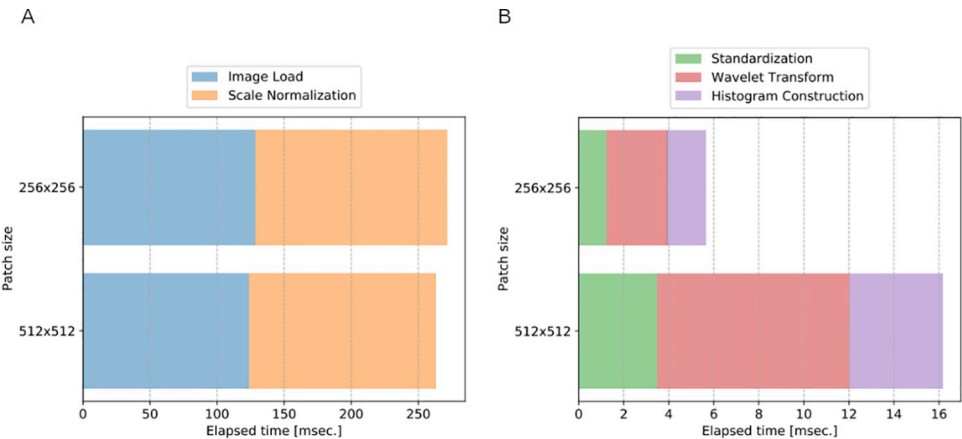

**Fig 3. The elapsed time of the image processing steps [msec.].** (A) The elapsed time of pre-processing. (B) The elapsed time of feature extraction.

and *Wavelet-3* are embedded into inseparable clusters (Fig 4A and 4B). On the other hand, *HoW-3* are embedded into clearly isolated clusters as expected (Fig 4C). These plots indicate that there are clear boundaries exist between positive and negative MAT images in *HoW* feature space.

For further investigation, Fig 5 compares *HoW* features of the training data with patch size $256 \times 256$. The shape of *HoW-0* varies even in the same category while *HoW-4* seems to have a unique shape for each category. This indicates that SVM well-classifies higher level *HoW*s but mis-diagnoses lower-level *HoW*s.

## Quantitative evaluation of MAT image feature classification

The third experiment is a quantitative evaluation of the MAT image feature classification. In this experiment, we conducted an image classification experiment with K-fold cross validation to quantitatively evaluate the image features.

For SVM, we use radial basis function as kernel and potential hyperparameters which are described as below:

$$C \in \{10^i | i = 0, 1, 2, 3\} \text{ and } \gamma \in \{10^i | i = -3, -2, -1, 0\}.$$

For K-fold cross validation, we use 60% of data for training and the remaining 40% for test data. The number of folds $K$ is set to 5. As evaluation criteria, we use Matthews Correlation

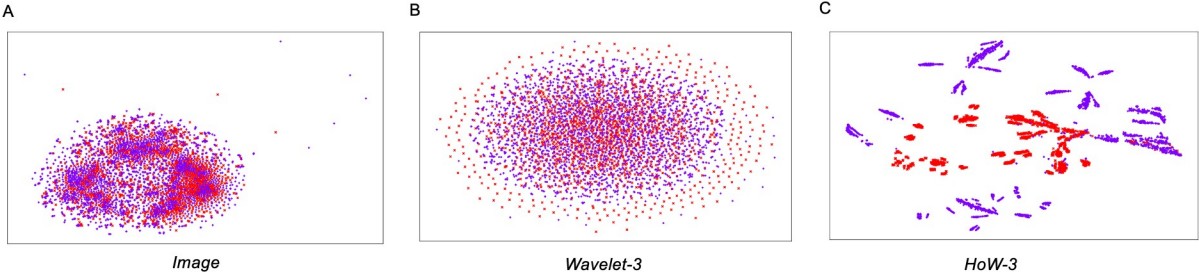

**Fig 4. The t-SNE 2D embedding of the features at the resolution of $256 \times 256$.** (A) *Image*, (B) *Wavelet-3*, and (C) *HoW-3*. Red (x) and purple (+) symbols represent the features of negative and positive patches respectively.

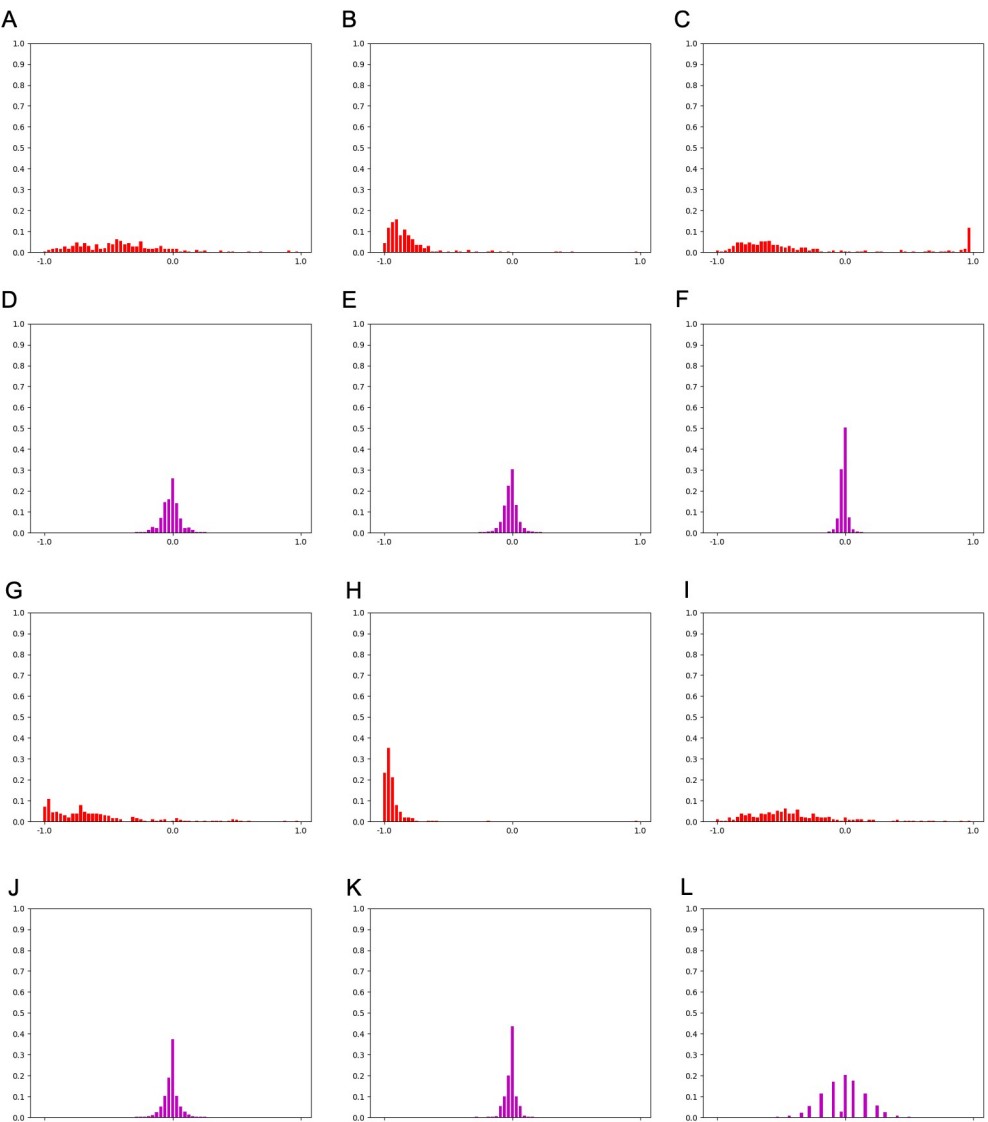

**Fig 5. A comparison of *HoW-0* and *HoW-4* features of patch size 256 × 256.** (A)-(C) *HoW-0* features of Negative data. (D)-(F) *HoW-4* features of Negative data. (G)-(I) *HoW-0* features of Positive data. (J)-(L) *HoW-4* features of Positive data.

Coefficient (MCC) [20] that ranges from -1 to 1, of which all predictions are wrong (value is -1), equivalent to random prediction (value is 0), or correct (value is 1). Contrary to F-measure, more frequently used criteria, MCC considers balance ratios of two classes that is crucial for our case.

Fig 6 shows confusion matrices and MCC values of each image feature with different patch sizes. Definitions of graph keys are as follows: True Negative, negative data is predicted as negative; False Positive, negative data is predicted as positive; False Negative, positive data is predicted as negative; True Positive, positive data is predicted as positive. In the figure, horizontal bars with (\) symbol represent true prediction while ones with (/) symbol indicate false prediction. In both resolutions, higher-level *HoWs* result in higher MCC while lower-level *HoW*, Image, and *Wavelet* result in lower MCC.

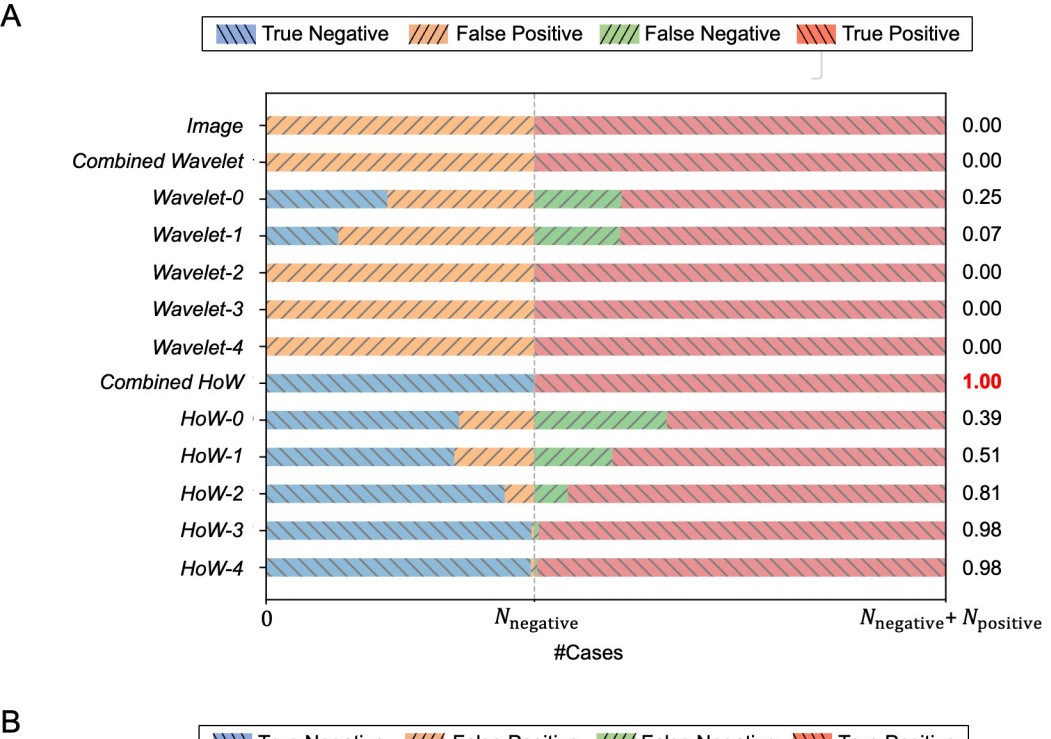

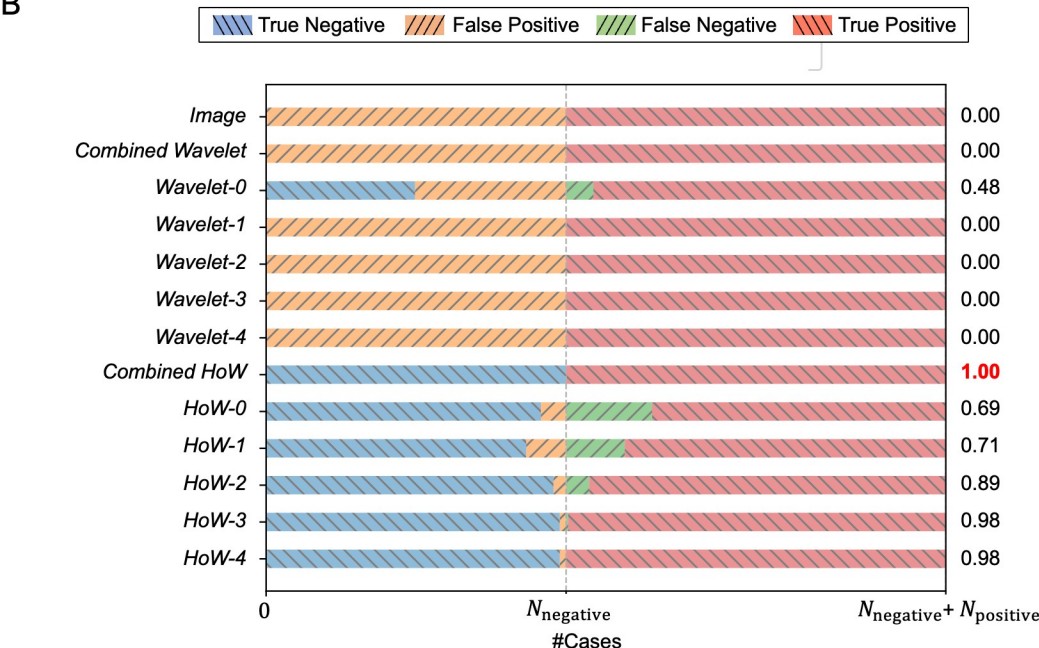

**Fig 6. Visualized confusion matrix and MCC for each feature of different patch sizes.** (A) Patch size $256 \times 256$. (B) Patch size $512 \times 512$.

Fig 7 shows representative succeeded or failed test cases of *HoW* features. Comparing those failed features to succeeded data or the training data shown in Fig 5, it is convincing that the SVM mis-classified those features.

Tables 5 and 6 show the elapsed time for the training and test respectively. Note that the elapsed time of the test is measured per each patch while the one of the trainings is measured per all the training datasets. Comparing the data between patch sizes, the elapsed time of

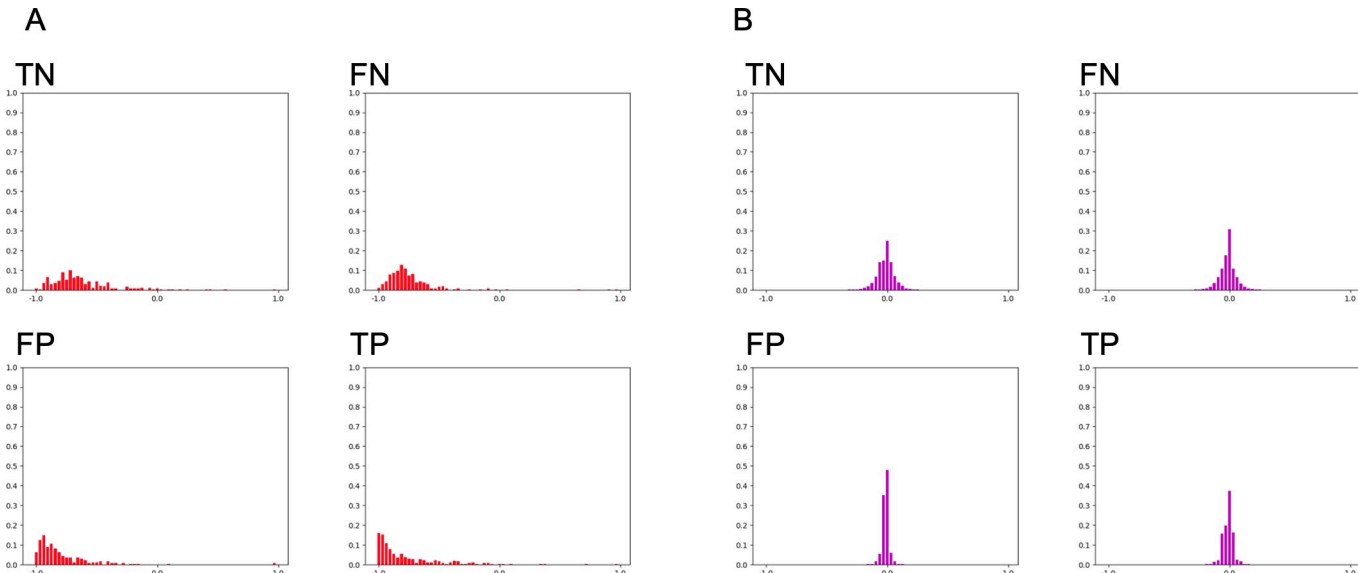

**Fig 7. A comparison of *HoW-0* and *HoW-4* features from the test cases of patch size 256 × 256.** Representative (A) *HoW-0* and (B) *HoW*-4 features of true negative (TN), false negative (FN), false positive (FP) and true positive (TP) cases.

256 × 256 cases is larger than 512 × 512 cases by 10 and 5 times on training and test cases, respectively. Considering the elapsed time and classification performance, we can say that HoW efficiently codes the characteristics of MAT images. This result indicates that SVM with higher-level *HoWs* work as a good MAT image classifier.

## Discussion

MAT is a standard test for diagnosis of leptospirosis and a determination training is required for clinical technologists, though there have still been variations among technicians or facilities. An alternative to MAT is the macroscopic slide agglutination test (MSAT), developed by Galton et al. [21]. The MSAT is considered as a rapid, practical, easy and accessible test, and is as sensitive as the MAT [22]. The MSAT was originally developed for the serological diagnosis

**Table 5. The elapsed time for training per dataset [sec.].**

|  | 256 × 256 | 512 × 512 |
|---|---|---|
| *Image* | $6.2 \times 10^3$ | $1.2 \times 10^3$ |
| *Combined-Wavelet* | $7.0 \times 10^3$ | $1.2 \times 10^3$ |
| *Wavelet-0* | $3.2 \times 10$ | $3.5$ |
| *Wavelet-1* | $8.5 \times 10$ | $1.0 \times 10$ |
| *Wavelet-2* | $3.0 \times 10^2$ | $4.0 \times 10$ |
| *Wavelet-3* | $1.2 \times 10^3$ | $1.6 \times 10^2$ |
| *Wavelet-4* | $4.6 \times 10^3$ | $7.0 \times 10^2$ |
| *Combined-HoW* | $1.0 \times 10$ | $6.7 \times 10^{-1}$ |
| *HoW-0* | $5.9$ | $3.4 \times 10^{-1}$ |
| *HoW-1* | $5.5$ | $3.2 \times 10^{-1}$ |
| *HoW-2* | $5.4$ | $3.6 \times 10^{-1}$ |
| *HoW-3* | $3.4$ | $2.9 \times 10^{-1}$ |
| *HoW-4* | $3.0$ | $2.7 \times 10^{-1}$ |

**Table 6. The elapsed time for test per patch [msec.].**

|  | $256 \times 256$ | $512 \times 512$ |
|---|---:|---:|
| *Image* | $7.9 \times 10^2$ | $1.5 \times 10^2$ |
| *Combined-Wavelet* | $8.6 \times 10^2$ | $1.5 \times 10^2$ |
| *Wavelet-0* | 2.7 | $4.9 \times 10^{-1}$ |
| *Wavelet-1* | 9.2 | 1.6 |
| *Wavelet-2* | $3.5 \times 10$ | 6.5 |
| *Wavelet-3* | $1.4 \times 10^2$ | $2.6 \times 10$ |
| *Wavelet-4* | $5.5 \times 10^2$ | $1.1 \times 10^2$ |
| *Combined-HoW* | $1.3 \times 10^{-1}$ | $1.6 \times 10^{-2}$ |
| *HoW-0* | $5.5 \times 10^{-1}$ | $1.9 \times 10^{-2}$ |
| *HoW-1* | $4.5 \times 10^{-1}$ | $1.6 \times 10^{-2}$ |
| *HoW-2* | $3.2 \times 10^{-1}$ | $1.6 \times 10^{-2}$ |
| *HoW-3* | $5.0 \times 10^{-2}$ | $4.0 \times 10^{-3}$ |
| *HoW-4* | $2.9 \times 10^{-2}$ | $5.4 \times 10^{-3}$ |

of leptospirosis in humans, mainly for the screening of acute and recent cases of infection [23]. The MSAT suggests the possible infective serovars using an antigen in suspension, which may include a pool of up to three inactivated serovars. The MSAT can be used for the diagnosis of leptospirosis in both humans [24, 25] and animals [26–28]. Other serological assays, such as the enzyme-linked immunosorbent assay (ELISA) [29, 30] can also be used to detect infection. However, these alternative tests are for screening, whereas MAT is needed for precise serotype diagnosis in areas of high prevalence. We aim to automate and standardize MAT, the conventional and standard diagnostic method, rather than developing a new method to replace MAT. We propose to introduce a machine learning approach for determination of antibody titer.

In this study, we attempted to standardize MAT by automating the determination process. Our experimental results showed that the proposed machine learning-based pipeline with the derived image feature well-recognized the agglutination in MAT images in a good precision, especially with combined *HoW* which resulted in the MCC score of 1.0 (a perfect prediction). This indicates that the proposed method could substitute knowledge and experience of the skilled examiner by using machine learning techniques. As we showed in the current paper, addition of parameters progresses capability of analysis, further tuning of the image feature may improve the classification performance.

Some spaces f improvements still remain in the proposed method. The standard MAT examination procedure determines positivity by comparing test MAT images to a reference image, while the proposed method classifies each MAT image based on the amount of agglutination areas within. To refine the proposed method, we need to establish an appropriate way to compare reference and MAT images in image feature space. We plan to apply several histogram comparison methods and select one with the best classification performance. Or we will design another image feature that shows a clear difference between positive and negative data when we compare reference and MAT images.

The limitation of this study is that we tested only one serotype condition, serovar Manilae, with a positive and a negative serum with an image acquisition device. Although we can conclude at least that the direction of the image analysis has been adequately investigated, we need to study and validate with data from other serovars and devices in the future.

The current study suggests that MAT will be fully automated in the future. Standard MAT procedure requires to transfer samples from 96-well plates to glass slides, and this is the most laborsome handling in all processes. However, if we automate all the manual procedures in

addition to the image analysis algorithm presented in this paper, it is possible to develop a quick MAT method.

We picture the final product of the proposed method as a cloud-based system. By launching this kind of system on the cloud, anyone can utilize the system via the internet and therefore we can support people in poor resource situations. Moreover, the cloud-based system will lead to collect a large amount of data from all over the world that will progress diagnosis of diseases further.

## Conclusion

This paper aimed to build a machine learning model on MAT as our first step toward the ultimate goal to automate the MAT procedures for the diagnosis of leptospirosis. Our idea was to introduce a typical machine learning-based image classification pipeline that represents images by an appropriate feature and uses an SVM to classify each MAT image based on the difference in the feature space. The conducted experiments validated that *HoW* is an efficient and effective feature for MAT image classification. From this evidence, we concluded that the machine learning-based image classification pipeline has a potential power of fully automated MAT which we are in the process of developing.

## Supporting information

**S1 File. Supporting information on machine learning.**
(DOCX)

**S1 Fig. An example of 2-dimensional non-linearly separable and non-separable data for binary classification.** Red circle and blue cross symbols represent negative and positive data. Curved lines represent boundaries obtained from training data of which green lines represent the best boundary. (A) A good training data. (B) Potential boundaries obtained from (A). (C) The best boundary obtained from (A). (D) A bad training data. (E) Potential boundaries obtained from (B). (F) The best boundary obtained from (B).
(TIF)

**S2 Fig. The data sequence of the pre-processing.** (A) Raw MAT images of Negative data. (B) Scale normalized images of Negative data. (C) Extracted patches of Negative data. (D) Raw MAT images of Positive data. (E) Scale normalized images of Positive data. (F) Extracted patches of Positive data.
(TIF)

**S3 Fig. An example of the scale normalization.** Yellow and red scale bars represent 50 and 20 micrometers respectively. (A) A raw MAT image of Negative data. (B) Image (A) with scale normalized. (C) A raw MAT image of Positive data. (D) Image (C) with scale normalized.
(TIF)

**S4 Fig. The concept of K-fold cross validation.**
(TIF)

**S5 Fig. The t-SNE 2D embedding of various features.** Red (x) and purple (+) symbols represent the features of negative and positive patches, respectively.
(TIF)

## Acknowledgments

We are grateful to Mr. Shoji Tokunaga for his advice on machine learning methodology.

## Author Contributions

**Conceptualization:** Yuji Oyamada, Ryo Ozuru.

**Data curation:** Yuji Oyamada, Ryo Ozuru.

**Formal analysis:** Yuji Oyamada, Ryo Ozuru, Toshiyuki Masuzawa, Satoshi Miyahara, Yasu-hiko Nikaido, Mitsumasa Saito, Sharon Yvette Angelina M. Villanueva.

**Funding acquisition:** Yuji Oyamada, Ryo Ozuru, Jun Fujii.

**Investigation:** Yuji Oyamada, Ryo Ozuru.

**Methodology:** Yuji Oyamada, Ryo Ozuru.

**Project administration:** Ryo Ozuru.

**Resources:** Toshiyuki Masuzawa, Satoshi Miyahara, Yasuhiko Nikaido, Mitsumasa Saito, Sha-ron Yvette Angelina M. Villanueva.

**Software:** Yuji Oyamada.

**Supervision:** Jun Fujii.

**Validation:** Yuji Oyamada, Ryo Ozuru, Satoshi Miyahara, Yasuhiko Nikaido, Fumiko Obata, Mitsumasa Saito, Sharon Yvette Angelina M. Villanueva.

**Visualization:** Yuji Oyamada, Toshiyuki Masuzawa.

**Writing – original draft:** Yuji Oyamada, Ryo Ozuru.

**Writing – review & editing:** Yuji Oyamada, Ryo Ozuru, Toshiyuki Masuzawa, Satoshi Miya-hara, Yasuhiko Nikaido, Fumiko Obata, Mitsumasa Saito, Sharon Yvette Angelina M. Villa-nueva, Jun Fujii.

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
