## [Decision Letter · Decision Letter 0]

17 Sep 2021

PONE-D-21-20745A Machine Learning Model of Microscopic Agglutination Test for Diagnosis of LeptospirosisPLOS ONE

Dear Dr. Ozuru,

Thank you for submitting your manuscript to PLOS ONE. After careful consideration, we feel that it has merit but does not fully meet PLOS ONE’s publication criteria as it currently stands. Therefore, we invite you to submit a revised version of the manuscript that addresses the points raised during the review process.

ACADEMIC EDITOR:

Include a brief description about the machine learning and its advantage in the field of medicine. Appropriate literature back up is necessary to support some new descriptions.

We look forward to receiving your revised manuscript.

Kind regards,

Kalimuthusamy Natarajaseenivasan

Academic Editor

PLOS ONE

Journal Requirements:

This work was partially supported by JSPS KAKENHI Grant Number 18K16174 and 21K16320 to R.O., the discretionary fund of Tottori University President to Y.O. and R.O, and the Research Program of the International Platform for Dryland Research and Education, Tottori University to J.F. The funders had no role in study design, data collection and analysis, decision to publish, or preparation of the manuscript.

Reviewers' comments:

Reviewer's Responses to Questions

**Comments to the Author**

1. Is the manuscript technically sound, and do the data support the conclusions?

Reviewer #1: Yes

Reviewer #2: Yes

2. Has the statistical analysis been performed appropriately and rigorously? 

Reviewer #1: N/A

Reviewer #2: Yes

3. Have the authors made all data underlying the findings in their manuscript fully available?

Reviewer #1: Yes

Reviewer #2: Yes

4. Is the manuscript presented in an intelligible fashion and written in standard English?

Reviewer #1: No

Reviewer #2: Yes

5. Review Comments to the Author

Reviewer #1: In this manuscript by Oyamada et al., the authors established a machine learning model to improve the microscopic agglutination test for the diagnosis of leptospirosis, one of the most widespread zoonoses in the world. The number of leptospirosis cases is underestimated in many countries due to the technical difficulties of the current available methods, thus, the development of this new method will greatly contribute to increase the number of laboratory-confirmed leptospirosis cases and to know the true burden of this neglected infectious disease.

I have some comments that can be useful to improve the manuscript within the scope of PLOS ONE:

1. INTRODUCTION: The detailed explanation of the microscopic agglutination test (MAT) is correctly provided, however, the background of machine learning methods applied to diagnosis in medicine is not presented. Please provide a background, with the appropriate references, on how image analysis and machine learning techniques have contributed to the diagnosis of other diseases. Some examples are: https://doi.org/10.1016/j.cmpb.2018.05.034 doi: 10.3389/fonc.2021.608191.

2. Provide also in the Introduction the appropriate background and references for image classification techniques, including the one used in this study (binary) using support vector machines, so that readers outside the field can understand the methodology presented in this study. With the use of appropriate references, the Fig. 1 can be omitted or presented as a supporting information.

3. MATERIALS and METHODS: Please provide a flowchart of the pipeline developed in this study as a Figure as in Ref. doi: 10.3389/fonc.2021.608191. (Fig. 2).

4. With the use of a flowchart, I suggest to brush-up the section “Materials and Methods”. For example some phrases such as,”In the remainder of this part,……, will be explained step by step” (last lanes of Image features paragraph) can be deleted. Other redundant phrases can be also deleted.

5. I suggest that a selection of panels for the main results of this study might be presented as Figures and that any other information presented in the current version as Fig. might be presented as “Supplementary Information” (i.e. Figures 1~5). Some panels of Figure 7-19 can be combined in one figure. The comparison of related-panels in one Fig. (as presented in Fig. 22) will help readers to follow the results.

6. Is it possible to compare HoW-0 and HoW-4 features of true negative, false negative, true positive, false positive in one Figure?

Minor comments:

1. Please include line numbers in the Manuscript which will help the revision process.

2. Check spelling, i.e.: “grasses” should be “glasses” in the second paragraph of Introduction.

3. The reference (Boser et al.,1992) for SVM is mentioned in the paragraph of “Image feature classification”, however, is not added in the Reference section.

Reviewer #2: In this study, automatic identification algorithm of MAT was developed and the possibility of machine learning diagnosis of MAT images was showed. The manuscript is well written and results are interesting. The presented data in revised manuscript is completely supports the conclusion of this study. The authors should be commended for conducting this novel and unique study to automatize the cumbersome and time-consuming MAT procedure. The only limitation of the study is evaluation restricted with one serovar.

6. PLOS authors have the option to publish the peer review history of their article (what does this mean?). If published, this will include your full peer review and any attached files.

Reviewer #1: No

Reviewer #2: **Yes: **SUMAIYA K

---

## [Author Response · Author response to Decision Letter 0]

1 Oct 2021

Dear Dr. Natarajaseenivasan, Editor of PLOS ONE,

We would like to thank all the editor and reviewers very much for reviewing our manuscript. We would like to submit a revised version of our manuscript entitled “A Machine Learning Model of Microscopic Agglutination Test for Diagnosis of Leptospirosis”, manuscript number PONE-D-21-20745 with all modifications following reviewers’ advice. Our responses to reviewers’ comments are below.

ACADEMIC EDITOR:

Include a brief description about the machine learning and its advantage in the field of medicine. Appropriate literature back up is necessary to support some new descriptions.

We appreciate your decision regarding our manuscript. Following the advice of the editor and reviewers, a description of the significance of machine learning in medicine has been added to INTRODUCTION section (page 4, lines 79-80, page 5, lines 81-101). Accordingly, some references have been added (references 7-14). Moreover, Tables 5 “The elapsed time for training per dataset [sec.].” and 6 “The elapsed time for test per patch [msec.].”, which were referred but not contained in the manuscript were added at the end of RESULTS section. Additionally, we have removed paragraphs from the DISCUSSION section that are not needed in this paper. We trust that this revised manuscript will answer all the questions and be improved to fit PLOS ONE.

Reviewer #1: In this manuscript by Oyamada et al., the authors established a machine learning model to improve the microscopic agglutination test for the diagnosis of leptospirosis, one of the most widespread zoonoses in the world. The number of leptospirosis cases is underestimated in many countries due to the technical difficulties of the current available methods, thus, the development of this new method will greatly contribute to increase the number of laboratory-confirmed leptospirosis cases and to know the true burden of this neglected infectious disease.

I have some comments that can be useful to improve the manuscript within the scope of PLOS ONE:

We appreciate your detailed reviews and constructive comments. The manuscript has been revised as follows according to the directions.

1. INTRODUCTION: The detailed explanation of the microscopic agglutination test (MAT) is correctly provided, however, the background of machine learning methods applied to diagnosis in medicine is not presented. Please provide a background, with the appropriate references, on how image analysis and machine learning techniques have contributed to the diagnosis of other diseases. Some examples are: https://doi.org/10.1016/j.cmpb.2018.05.034 doi: 10.3389/fonc.2021.608191.

Descriptions of machine learning and its medical applications are added in lines 79-89. We also cited several papers as references 7-9.

2. Provide also in the Introduction the appropriate background and references for image classification techniques, including the one used in this study (binary) using support vector machines, so that readers outside the field can understand the methodology presented in this study. With the use of appropriate references, the Fig. 1 can be omitted or presented as a supporting information.

We added background explanations on image identification and classification in lines 90-101. New citations #11-14 have been added accordingly. Just in case, the original description of binary classification and Fig. 1 have been moved to Supporting information (S1 File and S1 Fig, respectively).

3. MATERIALS and METHODS: Please provide a flowchart of the pipeline developed in this study as a Figure as in Ref. doi: 10.3389/fonc.2021.608191. (Fig. 2).

We inserted a flowchart as Fig 1 as you suggested.

4. With the use of a flowchart, I suggest to brush-up the section “Materials and Methods”. For example some phrases such as,”In the remainder of this part,……, will be explained step by step” (last lanes of Image features paragraph) can be deleted. Other redundant phrases can be also deleted.

Based on the flowchart, we modified the section “Materials and Methods”. Specifically, the following points have been corrected.

- Deleted some redundant phrases such as you pointed out.

- Moved image processing methods to Supporting information (S1 File).

- Moved Table 1 and 2 to the end of “preparation of image datasets of MAT”

5. I suggest that a selection of panels for the main results of this study might be presented as Figures and that any other information presented in the current version as Fig. might be presented as “Supplementary Information” (i.e. Figures 1~5). Some panels of Figure 7-19 can be combined in one figure. The comparison of related-panels in one Fig. (as presented in Fig. 22) will help readers to follow the results.

We reorganized figures and tables. Concretely,

- Figs 1-4 are moved to Supporting information (S1-S4 Figs).

- Among Figs 7-19, Figs 7 (Raw images), 12 (Wavelet-3), and 18 (HoW-3) are reorganized as Fig 4, because we think it is important to compare them together. The rest have been combined and moved to Supporting information (S5 Fig).

6. Is it possible to compare HoW-0 and HoW-4 features of true negative, false negative, true positive, false positive in one Figure?

The comparison results of your proposal are now shown in Fig 7. 

Minor comments:

1. Please include line numbers in the Manuscript which will help the revision process.

We included line numbers in the manuscript.

2. Check spelling, i.e.: “grasses” should be “glasses” in the second paragraph of Introduction.

The following spelling corrections and proofreading of the manuscript have been made.

ABSTRACT

Line33: Leptospiras -> Leptospires

Line40: aggregation -> agglutination

Line42: images created -> images created

Line42: Leptospira- -> Leptospira-

INTRODUCTION

Line58: 70% occurring -> 70% is occurring

Line67: dilution agglutinating -> dilution which agglutinates

Line69: i.e. -> i.e.

Line70: personnel making -> personnel, thus making

Line71: grasses -> glasses

Line72: takes time -> time consuming

Lines74-75: partially because -> because

Lines75-76: MAT such as dark-field microscopes, objective lenses, illuminations and cameras, but -> MAT (dark-field microscopes, objective lenses, illuminations and cameras) vary, but also

Line77: at the -> among

Lines133-135: raw images because raw images are too complex and redundant, which is potentially badly distributed data. -> raw images, that are considered too complex, redundant and potentially badly distributed data.

Line135: interesting -> characteristic

Line136: a sort of numerical -> numerical

Line137: Image -> Appropriate image

Line138: which means each -> such that

Line139: takes an important role -> is important

Line141: machine learning -> a machine learning

Line147: a -> an

MATERIALS AND METHODS

Line152: that takes -> which predict

Line174: said University -> university

Line178: syrian -> Syrian

Line184: was -> were

Line186: slide glasses -> glass slides

Line187: Nikon -> Nikon, Tokyo, Japan

Lines188-189: Olympus, Tokyo, Japan

Lines249-250: objects such as dust that has strong -> objects as dust with strong

Line252: Specifically, the designed -> The designed

Line255: histogram of its coefficients -> histogram

Line292: Namely, lower-level -> Lower-level

Line293: larger-level -> higher-level

Line297: an image and its multi-level -> same microscopic field in various multi-level

Line299: levels, coarser -> levels, that coarser

Line300: lower level -> lower-level

Line300: finer in higher -> finer ones are in higher

Lines301-302: features are depicted -> features such as points, curves, and edges with large coefficient changes are depicted

Line317: agglutination -> agglutinated

Line320: histograms of different -> different

Line340: utilizes -> utilized

Line341: well-used -> the well-used

Line350: Using kernel trick -> Using a kernel trick

Line351: An SVM -> a SVM

Line361: utilizes -> utilized

Line363: hyperparameter -> the hyperparameter

RESULTS

Line394: for -> with

Line405: test -> tested

Line435: is fastened -> becomes faster

Lines438-439: system that -> system, in which

Line443: all the data -> all datasets

Lines458-459: red x and purple + symbols represent negative and positive data respectively and -> red (x) and purple (+) symbols represent negative and positive data, respectively, and

Line463: distinctly -> clearly

Lines471-472: Red x and purple + symbol represent the feature -> Red (x) and purple (+) symbols represent the features

Line524: lower level -> lower-level

Lines539-541: from -1 to 1, -1, where -1 means all predictions are wrong, 0 means the predictions are equivalent to random prediction, and 1 means all predictions are correct. -> from -1 to 1, of which all predictions are wrong (value is -1), equivalent to random prediction (value is 0), or correct (value is 1).

Line549: \\ -> (\\)

Line550: / -> (/)

Line550: do -> indicate

Line567: training -> trainings

Line568: dataset -> datasets

DISCUSSION

Line589: considered -> considered as

Lines596-598: However, these alternatives are just simple rapid screening tests, and MAT is needed as ever in high-density areas where precisely testing with more serovars. -> However, these alternative tests are for screening, whereas MAT is needed for precise serotype diagnosis in areas of high prevalence.

Line603: primarily attempted -> attempted

Lines606-607: MAT images, especially combined HoW resulted in the MCC score of 1.0, which means perfect prediction. -> MAT images in a good precision, especially with combined HoW which resulted in the MCC score of 1.0 (a perfect prediction).

Lines608-609: the skilled examiners’ knowledge and experience -> nowledge and experience of the skilled examiner

Line610: adding parameters progress capability -> addition of parameters progresses capability

Lines612-613: There still remains some space to improve the proposed method. -> Some spaces f improvements still remain in the proposed method.

Lines613-614: procedure by comparing MAT images -> procedure determines positivity by comparing test MAT images

Line621-626: Although a lot of image data was acquired as a result, the limitation of this study is that we tested only one serotype condition, serovar Manilae, with a positive and a negative serum, and an image acquisition device. We need to study and validate with data from other serovars and devices but conclude at least that the direction of the image analysis has been adequately investigated. -> The limitation of this study is that we tested only one serotype condition, serovar Manilae, with a positive and a negative serum with an image acquisition device. Although we can conclude at least that the direction of the image analysis has been adequately investigated, we need to study and validate with data from other serovars and devices in the future.

Line636: Usually, MAT requires -> Standard MAT procedure requires

Line637: complicated -> laborsome

Line638-639: procedures, -> procedures in addition to the image analysis algorithm presented in this paper,

Line642-644: Furthermore, we can collect a huge amount of data from all over the world that is an image of the future research style. -> Moreover, the cloud-based system will lead to collect a large amount of data from all over the world that will progress diagnosis of diseases further.

CONCLUSION

Lines647-648: toward ultimate -> toward the ultimate

Line648: Leptospirosis -> leptospirosis

Line652: HoW features -> HoW

Line652: these -> this

Line654: in process -> in the process

3. The reference (Boser et al.,1992) for SVM is mentioned in the paragraph of “Image feature classification”, however, is not added in the Reference section.

Boser et al. 1992 is cited as #17 in the revised manuscript.

Reviewer #2: In this study, automatic identification algorithm of MAT was developed and the possibility of machine learning diagnosis of MAT images was showed. The manuscript is well written and results are interesting. The presented data in revised manuscript is completely supports the conclusion of this study. The authors should be commended for conducting this novel and unique study to automatize the cumbersome and time-consuming MAT procedure. The only limitation of the study is evaluation restricted with one serovar.

Thank you for identifying critical points of our study. We appreciate your comments. We also understand importance of evaluating the algorithm with other serovars and actual clinical samples, therefore we are planning future experiments utilizing the WHO panel and clinical samples in the coming project.

---

## [Decision Letter · Decision Letter 1]

29 Oct 2021

A Machine Learning Model of Microscopic Agglutination Test for Diagnosis of Leptospirosis

PONE-D-21-20745R1

Dear Dr. Ozuru,

We’re pleased to inform you that your manuscript has been judged scientifically suitable for publication and will be formally accepted for publication once it meets all outstanding technical requirements.

Kind regards,

Kalimuthusamy Natarajaseenivasan

Academic Editor

PLOS ONE

Additional Editor Comments (optional):

Reviewers' comments:

Reviewer's Responses to Questions

**Comments to the Author**

1. If the authors have adequately addressed your comments raised in a previous round of review and you feel that this manuscript is now acceptable for publication, you may indicate that here to bypass the “Comments to the Author” section, enter your conflict of interest statement in the “Confidential to Editor” section, and submit your "Accept" recommendation.

Reviewer #1: All comments have been addressed

Reviewer #2: All comments have been addressed

2. Is the manuscript technically sound, and do the data support the conclusions?

Reviewer #1: (No Response)

Reviewer #2: Yes

3. Has the statistical analysis been performed appropriately and rigorously? 

Reviewer #1: (No Response)

Reviewer #2: Yes

4. Have the authors made all data underlying the findings in their manuscript fully available?

Reviewer #1: (No Response)

Reviewer #2: Yes

5. Is the manuscript presented in an intelligible fashion and written in standard English?

Reviewer #1: (No Response)

Reviewer #2: Yes

6. Review Comments to the Author

Reviewer #1: (No Response)

Reviewer #2: (No Response)

7. PLOS authors have the option to publish the peer review history of their article (what does this mean?). If published, this will include your full peer review and any attached files.

Reviewer #1: No

Reviewer #2: **Yes: **SUMAIYA K

---

## [Editor Report · Acceptance letter]

6 Nov 2021

PONE-D-21-20745R1 

A Machine Learning Model of Microscopic Agglutination Test for Diagnosis of Leptospirosis 

Dear Dr. Ozuru:

I'm pleased to inform you that your manuscript has been deemed suitable for publication in PLOS ONE. Congratulations! Your manuscript is now with our production department. 

Kind regards, 

on behalf of

Dr. Kalimuthusamy Natarajaseenivasan 

Academic Editor

PLOS ONE